# *Stomoxys* Species Richness and Apparent Densities at Different Land-Use Setups in North-Eastern KwaZulu-Natal Province, South Africa

**DOI:** 10.3390/insects16101049

**Published:** 2025-10-15

**Authors:** Percy Moyaba, Serero Abiot Modise, Johan Esterhuizen, Keisuke Suganuma, Noboru Inoue, Oriel Thekisoe, Moeti Oriel Taioe

**Affiliations:** 1Agricultural Research Council—Onderstepoort Veterinary Research, Soutpan Road (M35), Onderstepoort 0110, South Africa; esterhuizenj@arc.agric.za (J.E.); taioem@arc.agric.za (M.O.T.); 2Unit for Environmental Sciences and Management, North-West University, Potchefstroom 2531, South Africa; oriel.thekisoe@nwu.ac.za; 3Foundational Biodiversity Science, South African National Biodiversity Institute, Pretoria 0001, South Africa; s.modise@sanbi.org.za; 4National Research Center for Protozoan Diseases, Obihiro University of Agriculture and Veterinary Medicine, Inada, Obihiro 080-8555, Japan; k.suganuma@obihiro.ac.jp (K.S.); ircpmi@obihiro.ac.jp (N.I.)

**Keywords:** *Stomoxys* species, apparent densities (ADs), north-eastern KwaZulu-Natal

## Abstract

The genus *Stomoxys* contains about 18 species recognized worldwide, and they are parasitic blood-feeding flies of medical and veterinary importance. There is a scarcity of well-documented information regarding the abundance and diversity of these flies in the north-eastern KwaZulu-Natal province (KZN) of South Africa. A total of 10 localities divided into three land-use setups, namely communal farming areas, commercial farms, and private game farms, were sampled using odor-baited H-traps. This study identified six collected *Stomoxys* species, namely *Stomoxys calcitrans* (Linnaeus, 1758), *S niger niger* (Macquart, 1851), *S. sitiens* (Rondani, 1873), *S. taeniatus* (Bigot, 1888), *S. n. bilineatus* (Grunberg, 1906), and *S. boueti* (Roubaud, 1911), which co-exist with tsetse flies, the biological vectors of *Trypanosoma* parasites in the sampled area. Among the six species, *S. n. niger* was the most abundant species captured from all the sampled sites. The presence of these species constitutes a significant animal health risk due to their vectorial role in the transmission of various disease-causing pathogens of medical and veterinary significance. Future vector control campaigns in KZN should not be limited to ticks, mosquitoes, and tsetse flies, but should be extended to other biting and blood-feeding insects in the area, including *Stomoxys* species.

## 1. Introduction

*Stomoxys* (Muscidae: *Stomoxyinae*) is a genus of hematophagous flies with a cosmopolitan distribution associated with livestock, wild animals, and occasionally humans as hosts [1]. Fourteen of the 18 known species of *Stomoxys* Geoffroy (1762) occur on the African continent [2]. Both adult sexes feed exclusively on blood and can, as such, inflict painful bites on their hosts, leading to significant blood losses [3,4]. *Stomoxys* flies are not only nuisance pest insects but have also been implicated as biological or mechanical vectors of several infectious pathogens globally, including bacteria, viruses, protozoa, and various helminths [5,6]. Apart from transmission of pathogens during blood feeding, their annoying biting behavior causes stress to animals, leading to changes in feeding patterns and overall poor performance of the affected farm. Economic losses caused by *Stomoxys* flies in the United States alone are estimated at over 1 billion US dollars annually [4]. The severe biting by these flies makes them an important pest of cattle, substantially leading to weight loss and reduced milk production.

*Stomoxys* flies are distributed across a variety of areas ranging from boreal, tropical, and temperate climatic zones, with individual species highly variable in terms of composition [7], abundance [8], and habitat choice, depending on the surrounding local environmental conditions. These flies have vast biotope preferences, including forests, livestock breeding sites, or game farms, and in most areas are associated with human and animal husbandry activities [5,9]. The flies have adapted to areas where warm-blooded animals congregate with a routine behavioral pattern and increase chances of feeding and larval survival rates [10]. These include stables, slaughterhouses, cattle markets, feedlots, dairy farms, rubbish dumps, and other favorable habitats with organic matter [11,12]. Furthermore, their presence has been documented from other habitats such as the savannah, shrubland, grassland, coastal, riverine, and semi-arid areas [13,14,15], with ecological attributes to feed and breed along with both domesticated and wildlife animals in their vast surrounding environments [2].

Environmental changes due to anthropogenic activities and land use have been shown to affect the vector and feeding host relationship community dynamics by habitat alteration [16] and may influence species composition [13] and increase or decrease in larval developmental sites [16]. Some studies have indicated that the effect of land use might indirectly and directly influence Stomoxyine species composition [17].

In South Africa, recent studies focused only on *S. calcitrans* (Linnaeus, 1758), which is commonly due to its cosmopolitan distribution [18,19,20]. A study by Makhahlela et al. [21] reported the occurrence of *S. calcitrans* from feedlots in three provinces: Free State, Limpopo, and North West, and further detected the presence of *Anaplasma marginale* and Lumpy skin disease virus DNA from this fly species. Similarly, Evert [19] reported *S. calcitrans* to be the only species found in abundance at a feedlot in Gauteng Province of South Africa during her study period. There is a scarcity of detailed information relating to the species diversity within the genus *Stomoxys* in South Africa, specifically in the north-eastern KwaZulu-Natal Province. This area is of interest because it is a historical animal trypanosomosis (Nagana) focus and forms the southernmost distribution of tsetse flies (*Glossina* spp.) on the African continent [22,23,24]. Research has shown that *Stomoxys* flies can mechanically transmit trypanosome parasites [5], making it essential to determine their abundance, diversity, and ecology due to their role as biological and mechanical vectors of disease-causing pathogens of medical and veterinary importance. Therefore, this study sought to determine the *Stomoxys* species diversity and richness, and the apparent density (AD) from north-eastern KZN Province in three different land-use setups, namely communal farming areas, commercial farms, and private game farms, in an effort to contribute knowledge on the diversity of potential mechanical vectors of animal trypanosomosis in KZN Province.

## 2. Materials and Methods

### 2.1. Study Area

Subsistence farming is the most common farming practice in north-eastern KZN Province with numerous communal farms interspersed with several protected areas such as the provincial game parks, private game parks, and reserves [25] (Figure 1). The sampled areas form part of the Greater St Lucia Wetlands Park, a World Heritage Site where insect diversity is of conservation interest. The sampled areas were selected based on their proximity to protected areas, where there is a possible livestock–wildlife interface and previous historical tsetse infestation records. This area falls under the Savanna biome, composed of structural vegetation types ranging from sandy bushvelds, clay bushvelds, and coastal thornvelds [26]. Additionally, there are patches of wooded grassland that consist of plantations of exotic *Eucalyptus* sp. and *Pinus* sp. trees, which are mainly used for wood and paper [27]. These areas have frequent rains in the summer with annual mean precipitation ranging between 600 mm and 1050 mm [26]. The traps were set up in 10 localities divided into three land-use setups: communal farmlands, commercial farms, and private game farms (wildlife) (Figure 1).

### 2.2. Fly Collection and Study Design

This study is part of the COMBAT project [28] and the results form part of the second work package (WP2) on the role of mechanical vectors in the transmission of trypanosome parasites. These samples were opportunistically captured during targeted surveys for tsetse flies throughout selected areas across the north-eastern KwaZulu-Natal Province. Field flies were captured using odor-baited H-traps (Appendix A Appendix A) developed for capturing tsetse flies in South Africa [29]. In total, thirty-four traps were deployed, with a minimum of 2 traps per site placed at a minimum distance of at least 100 m apart, to avoid cross-attraction between traps. Trap ID, location name, and GPS coordinates were recorded for each trap, and the traps were deployed for 30 days during the autumn season (March 2023). All the traps were baited with 1-octen-3-ol and 4-methylphenol at 1:8 ratio, released at 4.4 mg/h and 7.6 mg/h [25]. The chemicals were dispensed from eight heat-sealed sachets made of low-density polyethylene sleeves (7 cm × 9 cm) placed at the entrance of each trap. A 300 mL brown glass bottle was placed at the entrance of each trap and dispensed acetone through a 6 mm hole from the lid, as shown in Appendix A Appendix A. The traps consisted of two plastic collection bottles, which were half-filled with 70% ethanol and disinfectant mixture to preserve all the collected specimens, to prevent predator insects from eating collected specimens, and to prevent fecal DNA cross-contamination among collected insects [30]. The traps were serviced daily, and collections were transferred into storage bottles. The captured flies were brought to the Agricultural Research Council—Kuleni field station and preserved in 70% ethanol, and the date of collection, trap number, and location ID were recorded on the label until the number and identification of specimens were recorded.

### 2.3. Morphological Identification of Stomoxys spp.

All the specimens were subsequently brought back to the Agricultural Research Council—Onderstepoort Veterinary Research, Entomology Unit in Pretoria for morphological identification using the Stemi 305 Trino with Axiocam 208 color camera (Carl Zeiss, Oberkochen, Germany), dissecting microscope (Carl Zeiss, Oberkochen, Germany), and the key of Zumpt (1973). Features such as the color of the thorax, width of the frons in males, dorsal abdominal patterns, and genitalia of the fly were used as distinguishing features for each species. For each species collected, 2 samples were kept as voucher specimens, and images of each specimen were taken as reference (Appendix A Appendix A). The number of flies, species, sex, and the place and date of collection were recorded. Each fly was placed individually in labeled 1.5 mL microcentrifuge tubes and stored at −20 °C until used for downstream molecular analysis.

### 2.4. Molecular Identification of Stomoxys spp.

#### 2.4.1. Genomic DNA Extraction 

The total genomic DNA was extracted from the whole body of an individual adult fly using the DNeasy Blood and Tissue Kit (Qiagen, Hilden, Germany) according to the manufacturer’s protocol to ensure maximum accuracy of the procedure. The last step, DNA was eluted at the final volume of 100 µL and stored at −20 °C until PCR analyses.

#### 2.4.2. PCR Amplification and Sequencing

The isolated gDNA was amplified by PCR targeting the 650 pb mitochondrial cytochrome oxidase I (COI) gene using the universal primers by Folmer et al. [31]. The PCR was performed in a thermocycler (BioRad, Berkley, CA, USA) with a total reaction volume of 25 µL, containing 12.5 µL of 1× One Taq Quick-load Master Mix [20 mM Tris-HCl, 1× Tartrazine, 25 units/mL OneTaq^®^ DNA Polymerase, 22 mM KCl, 22 mM NH_4_Cl, 1.8 mM MgCl_2_, 5% Glycerol, 0.06% IGEPAL^®^ CA-630, 0.05% Tween^®^ 20, 0.2 mM dNTPs, 1× Xylene Cyanol, pH 8.9 @ 25 °C] (NEB, Ipswich, MA, USA), 0.5 μM of each primer and 3 µL DNA. The PCR conditions were as follows: 94 °C for 2 min as initial denaturation step, 35 cycles of denaturation at 94 °C for 30 s, with the annealing temperature of 58 °C for 40 s, and initial extension step at 72 °C for 1 min, and the final extension performed at 72 °C for 10 min.

#### 2.4.3. Gel Electrophoresis and DNA Sequencing

Amplified PCR products were visualized under UV light on a 1% agarose gel stained with ethidium bromide, using the GelDoc Go Imaging System (BioRad, Berkeley, CA, USA). Subsequently, positively amplified DNA was purified using the QIAGEN Gel Purification Kit (Qiagen, Hilden, Germany) according to the manufacturer’s protocol. Sequencing PCR was performed using BigDye Terminator Cycle Sequencing Kit according to the manufacturer’s protocol (Applied Biosystems, Foster City, CA, USA). The eluent was loaded into the 96-well plate and placed in the ABI Prism 3100 Genetic Analyser, where sequencing electrophoresis took place using a 36 cm capillary array and POP 7 ^TM^ polymer (Applied Biosystems, Waltham, MA, USA). The generated sequences were visualized using the Sequence Analyzer software version v. 3.6 (Applied Biosystems, Waltham, MA, USA).

#### 2.4.4. Phylogenetic Analyses

The retrieved ABI sequences were edited for mixed base pairs and trimmed using TraceEditor on MEGA 12 [32]. Consensus sequences from the forward and reverse sequences were generated on the Alignment Explorer on MEGA 12 and saved as FASTA format. Subsequently, the Basic Local Alignment Search Tool for nucleotides (BLASTn) (BLAST; https://blast.ncbi.nlm.nih.gov/Blast.cgi (accessed on 19 April 2025)) was used to confirm the identity of the generated sequences to those deposited in GenBank from the NCBI database. Post confirmation, the generated sequences from this study were deposited into GenBank for accession.

The ClustalW algorithm was used to align congener sequences from the NCBI database and those generated from this study on the Multiple Alignment Explorer embedded in MEGA 12 [32]. Goodness of fit for multiple substation models using the Bayesian Information Criterion (*BIC*) scores was estimated for maximum likelihood analysis using 1000 bootstrap support values on MEGA12. The final tree topology with *Musca domestica* as the outgroup was visualized on the Tree Explorer in MEGA 12 [32].

### 2.5. Statistical Analysis

Microsoft Excel (2023) spreadsheet was used to enter the raw data for all statistical analyses. Percentages to represent confidence intervals of the mean at 95% were used to represent the total number of flies caught for each species.

The species surrogates were compared between three land-use categories (communal, commercial, and game farm), including methods of biotic indices to calculate relative abundance, and we estimated species richness using the Chao1 estimator [32], heterogeneity and evenness using the Shannon–Wiener index and Simpson–Yule function [33,34] at an alpha-sampled scale [35]. One-way ANOVA was used to determine the significance of the abundance of the different *Stomoxys* species sampled.

The apparent density (AD) was used to calculate the average number of flies caught per trap per day (flies/trap/day or FTD) over the sampled areas for each trapping site using the formula:FTD = ∑F/(T × D)(1)
where FTD is the flies/trap/day, ΣF is the total number of *Stomoxys* flies captured, T is the number of traps deployed per site, and D is the number of trapping days [36].

## 3. Results

### 3.1. Morphological Identification and Diversity

A total number of 1328 *Stomoxys* specimens comprising six species were collected; however, due to some specimens having damaged features, only 1306 specimens (828 males and 478 females) were used for the final analysis of the apparent densities (ADs). A total of 60 specimens comprising 10 individual specimens for each species were used to extract genomic DNA and subjected to PCR amplification targeting the cytochrome oxidase 1 (CO1) gene to supplement morphological identification. The collected specimens were morphologically identified as *S. calcitrans*, *S. n. niger*, *S. n. bilineatus*, *S. sitiens*, *S. taeniatus*, and *S. boueti*. (Appendix A Appendix A). The deposited voucher specimens were accessioned as OVIPC2024.1 (*S. sitiens*). OVIPC2024.3 (*S. n. niger*), OVIPC2024.5 (*S. calcitrans*), OVIPC2024.8 (*S. bilineatus*), OVIPC2024.10 (*S. taeniatus*), and OVIPC2024.12 (*S. boueti*). The most abundant species from the sampled sites was *S. n. niger* 82.3% (107.5 ± 89.5), followed by *S. calcitrans* 13.1% (17.1 ± 22.3), *S. n. bilineatus* 0.84% (1.1 ± 2.0), *S. taeniatus* 1.9% (2.5 ± 2.7), *S. sitiens* 1.1% (1.5 ± 2.7), and *S. boueti* 0.7% (0.9 ± 1.4) was the least collected. There was a significant difference in the overall abundance (*F* = 6.207; *p* = 0.0016, df = 3) of the collected *Stomoxys* species across the three different land-use setups.

### 3.2. Molecular Identification and Phylogenetic Analysis

From the 60 individual specimens used for CO1 amplification, only 26 samples were successful and generated sequences representing all morphologically identified *Stomoxys* species and were deposited into GenBank under accession numbers PV664306 to PV664329.

Maximum likelihood tree topology using the General Time Reversible with gamma distribution and invariable sites (GTR + G + I) (ln *L* = −1267.60) model of nucleotide substitutions based on the *BIC* showed a monophyletic clade within the Muscidae family (Figure 2). Additionally, paraphyly within the genus *Stomoxys* was confirmed, whereby a total of seven clades were observed, composed of *S. n. niger* clade (A), *S. bengalensis* clade (B), *S. taeniatus* clade (C), *S. sitiens* clades (D and F), *S. calcitrans* clade (E), and *S. n. bilineatus* clade (G). Generated sequences from this study were grouped with similar species deposited into GenBank with high bootstrap support values.

### 3.3. Species Surrogates

Species richness was higher in Zulucrock and Mkonge, with all six species recorded at the game farms, while lower in all the commercial and communal farm sites, recording only two species. Furthermore, higher species heterogeneity was also observed in the game farms, especially at Zulucrock (*H* = 0.300) and Mkonge (*H* = 0.181) sites, while the lowest was at Mvutshini (*H* = 0.221) and Ocilwane (*H* = 0.090) in the communal farms, respectively. Lastly, species composition differed from the three settings, with the Shannon–Wiener diversity index (*D* value) ranging between 0.57–0.950 and 0.100, respectively, but with a certain degree of variation in the communal farm sites (Figure 3). All the sampled sites shared common species of *S. n. niger* and *S. calcitrans*, while *S. taeniatus*, *S. niger bilineatus*, *S. sitiens*, *and S. boueti* were observed only in the game farm sites and commercial farms (Figure 4).

### 3.4. Apparent Densities (ADs) of Stomoxys Flies in the Sampled Land-Use Settings

The H-traps collected an overall (AD) of 1.28 (FTD) from the three land-use setups in this study area. *S. n. niger* and *S. calcitrans* were the most widely distributed species collected from all the sampled land-use setups (Figure 5). Higher AD of *Stomoxys* flies was observed at the private game farms compared to the commercial and communal farmlands. Mkonge game farm had the highest (AD = 9.75 FTD) for *S. n. niger*, followed by Zulucrock (AD = 8.15 FTD) and Boomerang commercial farm (AD = 0.61 FTD). Zulucrock recorded a high AD of 2.20 FTD for *S. calcitrans*, followed by Mkonge game farm (AD = 0.32).

For the communal lands, *S. n. niger* was also abundant with an AD of 0.86 at Ekuphindisweni, followed by Tembe communal farmland (AD = 0.36 FTD). However, Tembe recorded fewer *S. calcitrans* (AD = 0.02 FTD). *Stomoxys taeniatus* was the least collected from the game farms (AD = 0.21 FTD). No *S. calcitrans* was captured from the two communal areas Mvutshini and Ocilwane, while Ekuphindisweni recorded low numbers of this species.

From the sampled commercial farms, only *S. n. niger*, *S. calcitrans*, and *S. boueti* were collected, with Sukari feedlot recording the highest for both species as compared to Silversands and Boomerang commercial farms (Table 1).

A comparison of the ADs in the three settings showed no significant difference among the sampled sites (*p* = 0.857). *Stomoxys n. niger* had the highest AD = 10.79 (FTD) amongst all the collected species, followed by *S. calcitrans* at AD = 2.43. *Stomoxys n. bilineatus*, *S. taeniatus*, *S. sitiens*, *and S. boueti* were rarely caught, with AD = 0.61 (FTD) and AD = 0.43 FTD, respectively. The lowest AD was from the Ocilwane communal land with AD = 0.07 (FTD), with only *S*. *n. niger* and none of the other species collected (Table 1).

## 4. Discussion

The current study focused on determining the apparent density (AD) and species diversity and richness of *Stomoxys* flies found in three different land-use setups (commercial farms, communal farming lands, and private game farms) in north-eastern KwaZulu-Natal Province, where these flies co-exist with tsetse flies (*Glossina* spp.), the biological vectors of animal trypanosome parasites. In both morphological identification and molecular amplification of the COI gene, six species were revealed for the first time in this area, namely, *S. calcitrans*, *S. n. niger*, *S. n. bilineatus*, *S. sitiens*, *S. taeniatus*, and *S. boueti*. Of the six species, *S. n. niger* was the most abundant species across all the sampled sites, followed by *S. calcitrans* and *S. taeniatus*, whilst *S. sitiens*, *S. n. bilineatus*, and *S. boueti* were least detected.

The phylogenetic analysis based on the maximum likelihood method showed that CO1 gene sequences of *Stomoxys* species from the current study match with species sequences available in the GenBank, thereby confirming accurate species identification and that they form a paraphyletic clade within the genus. These findings correspond to those made by Dsouli et al. [11], where paraphyly in *Stomoxys* flies was observed from two mitochondrial gene markers (CO1 and CytB) and the nuclear ribosomal internal transcribed spacer 2 (ITS-2) gene. Additionally, the tree topology (Figure 2) is similar to that of Muita et al. [37], where two clades of *S. sitiens* are observed. In both studies, it is evident that the *S. sitiens* from Africa are genetically distant from those found in Asia. Lastly, the phylogenetic positions of *S. n. niger* and *S. n. bilineatus* suggest that they are evolutionarily diverse from each other. This corroborates observations made by Duvallet & Hogsette [2] that the subspecies *S. n. niger* and *S. n. bilineatus* are considered to be distant species from each other by molecular divergence and ecological preferences.

In South Africa, similar studies conducted in other provinces (Gauteng, Limpopo, and North West) reported presence of *S. calcitrans* species only from their respective sampled areas, which were cattle feedlots [18,19,20], whilst in the current study, *S. n. niger* was the most abundant and widely distributed species. Nonetheless, the current study is in accordance with the findings by Lendzele et al. [38], Bitome Essono et al. [14], Mavoungu et al. [39], Ahmed et al. [40], and Mihok & Clausen [41], where they reported *S. n. niger* being the most abundant species in other parts of Africa, followed by *S. calcitrans*. In the current study, *S. n. niger* were observed in all the sampled sites, but higher occurrences were recorded in game farms, suggesting that it might be more associated with wild animals than livestock. On the contrary, in Madagascar and the islands of La Réunion, *S. n. niger* occurred in large numbers close to the dairy cattle barns and showed variation in seasonal abundance among the regions [42].

*Stomoxys taeniatus* was first described in South Africa by Bigot in 1888 [2], and no other study in South Africa has ever reported its presence; this study is the first to report on this species, including the other documented species (*S. n. bilineatus*, *S. sitiens*, *S. n. niger*, and *S. boueti*) from South Africa. However, these species were in lower densities from the commercial and communal farmlands, suggesting that more investigations on *Stomoxys* flies in tsetse-infested areas are required, as observed in studies by Muita et al. [37] and Adungo et al. [43], where they reported the presence of *Stomoxys* flies in some tsetse-infested areas.

In this study, the apparent density and number of catches of *Stomoxys* differed between the trap locations and land-use setups. High apparent densities of *Stomoxys* flies were recorded from game and commercial farms where farming is intensive and there is availability of large number of suitable hosts and the abundance of suitable material to support their reproduction, such as animal droppings, which are the preferred egg-laying sites for these flies. In most cases, *S. calcitrans* and *S. n. niger* have been collected from the same sites in high numbers. This trend has been observed in almost all settings except in Ocilwane village, a communal farmland with the absence of *S. calcitrans*. The low frequency of *Stomoxys* flies in communal farmlands compared to the other settings can be explained by the open biotopes as secondary forest, savannah, or villages. In open areas, the abundance of *Stomoxys* spp. tends to decrease with distance from the livestock hosts [44]. The host density for blood-feeding adults and the suitable environmental conditions for larvae to complete their life cycle play a significant role in the abundance of these flies. This was seen in the current study, with high apparent densities observed at the private game farms due to the presence of various wildlife as potential blood meal sources. The current study shows that game farms are sites with the greatest diversity of *Stomoxys* flies, as exhibited by Mkonge and Zulucrock, which reported the presence of almost all six species in high abundance.

The use of tsetse traps to collect *Stomoxys* and other haematophagous flies is standard practice and is well-documented (e.g., Mihok [45], Esterhuizen [46]; Tunnakundacha et al. [44]). The ideal trap to be used for catching Stomoxyinae is the Vavoua trap [21,47,48]. However, H-traps were used in the current study, and they proved to be useful in catching *Stomoxys.* On the contrary, similar studies in South Africa that reported only on the presence of *S. calcitrans* used the Vavoua traps [20] or the Nzi traps [19].

The higher species diversity from the current study, compared to the previous works in South Africa, may be explained by the different environmental niches sampled. The previous studies were performed in commercial feedlots, while our study was conducted in tsetse infested areas of north-eastern KwaZulu-Natal and different ecological niches composed of variable land uses were sampled such as communal areas with invasive alien (*Acacia mearnsii*) and natural (*Vachellia karroo* and *V. xanthophloea*) vegetation, commercial farms some with feedlots near game reserves, and private game farms with wild animals. The presence of six different *Stomoxys* species in our study may indicate higher diversity in natural habitats rather than altered environments, such as feedlots. It is worth noting that the combination of the availability of host and habitat contributes to the abundance and diversity of *Stomoxyinae* flies.

## 5. Conclusions

This is the first study to reveal a variety of *Stomoxys* fly species occurring in the three land-use setups in the north-eastern KwaZulu-Natal. Six *Stomoxys* species (*S. n. niger*, *S. calcitrans*, *S. n. bilineatus*, *S. sitiens*, *S. taeniatus*, and *S. boueti*) were captured with an overall apparent density of 1.28 (FTD) using an H-trap designed for capturing tsetse flies. The data obtained from the current study provides insight into the abundance and distribution of other blood-feeding fly species cohabiting with tsetse flies. Given the uneven sampling for the north-eastern KZN region, other insects such as tsetse flies and tabanids, which are known to transmit pathogens in the region, were also collected and will be used in separate studies. Additional analysis to screen for protozoan pathogens and preferred mammalian hosts will be conducted on the collected *Stomoxys* samples, and the findings will highlight the actual role played by these flies in the transmission of protozoan pathogens, including trypanosomes.

## Figures and Tables

**Figure 1 insects-16-01049-f001:**
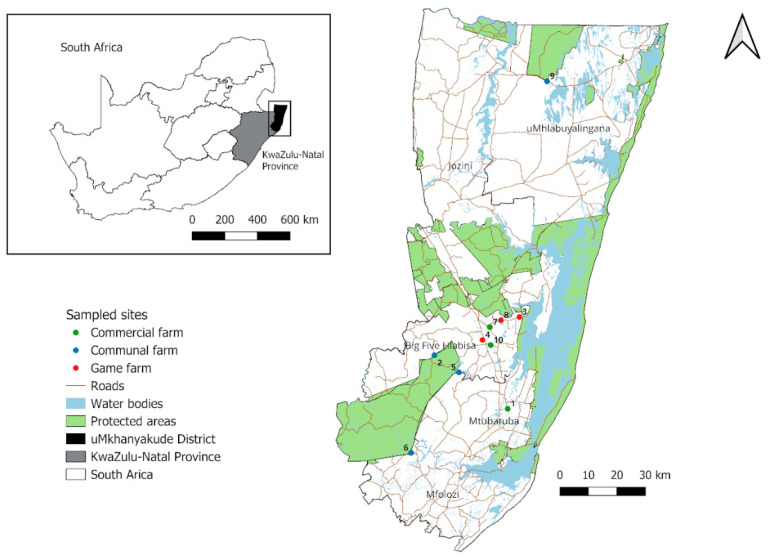
Sampled areas in the uMkhanyakude district of north-eastern KwaZulu-Natal Province with colored circles indicating three types of land use. (1 Boomerang, 2 Ekuphindisweni, 3 Kuleni, 4 Mkonge, 5 Mvutshini, 6 Ocilwane, 7 Silversands, 8 Zulucrock, 9 Tembe, and 10 Sukari).

**Figure 2 insects-16-01049-f002:**
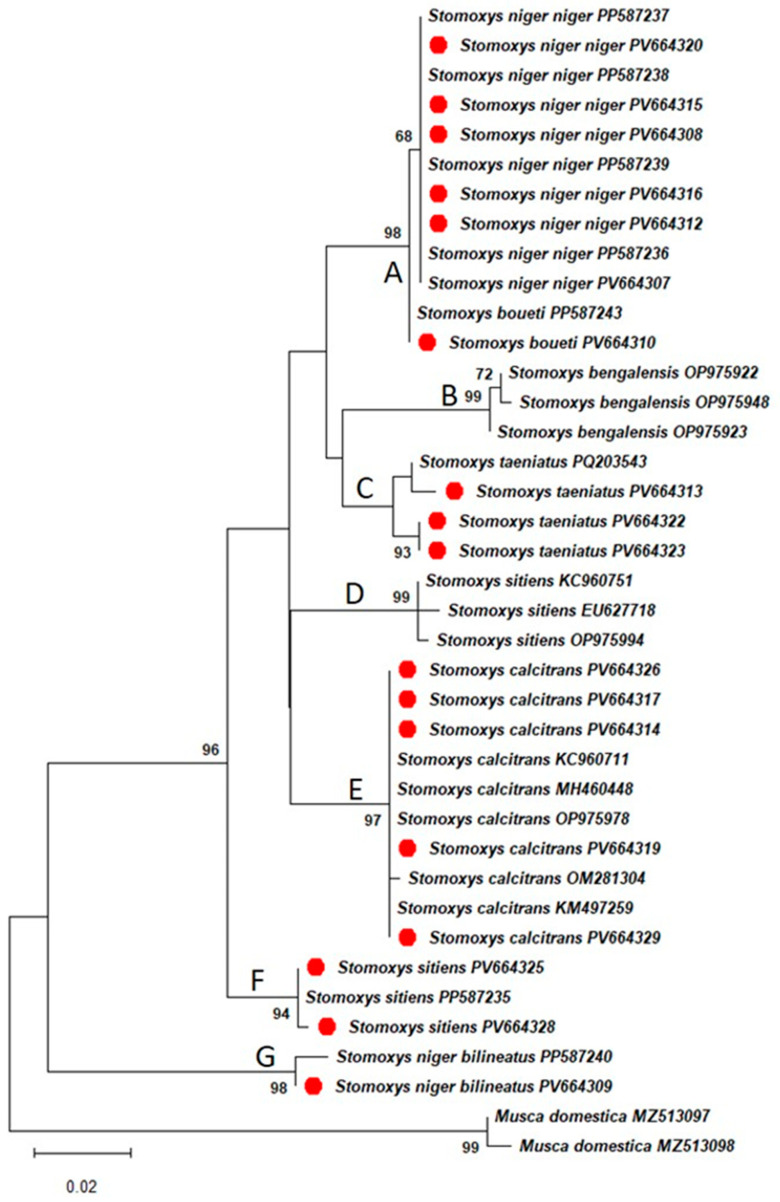
A maximum likelihood tree was constructed based on the GTR + G + I nucleotide substitution model for *Stomoxys* flies with *Musca domestica* as an outgroup, and samples from this study are represented in red color. The clades A-G represent: A: *S. n. niger* clade, B: *S. bengalensis* clade, C: *S. taeniatus* clade, D: *S. sitiens* clades from Asia, E: *S. calcitrans* clade, F: *S. sitiens* clades from Africa, and G: *S. n. bilineatus* clade.

**Figure 3 insects-16-01049-f003:**
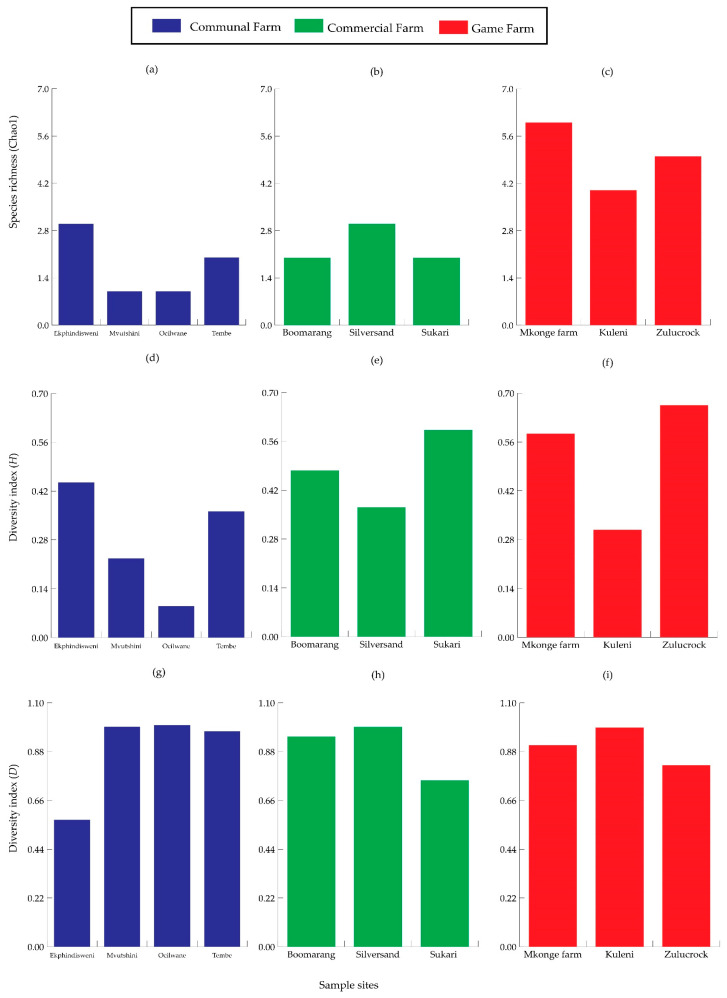
Species surrogates for *Stomoxys* composition comparing species richness (Chao1) for (**a**–**c**), heterogeneity (Shannon–Wiener diversity) for (**d**–**f**), and evenness (Simpson–Yule function) for (**g**–**i**) in communal, commercial, and game farms of north-eastern KwaZulu-Natal Province.

**Figure 4 insects-16-01049-f004:**
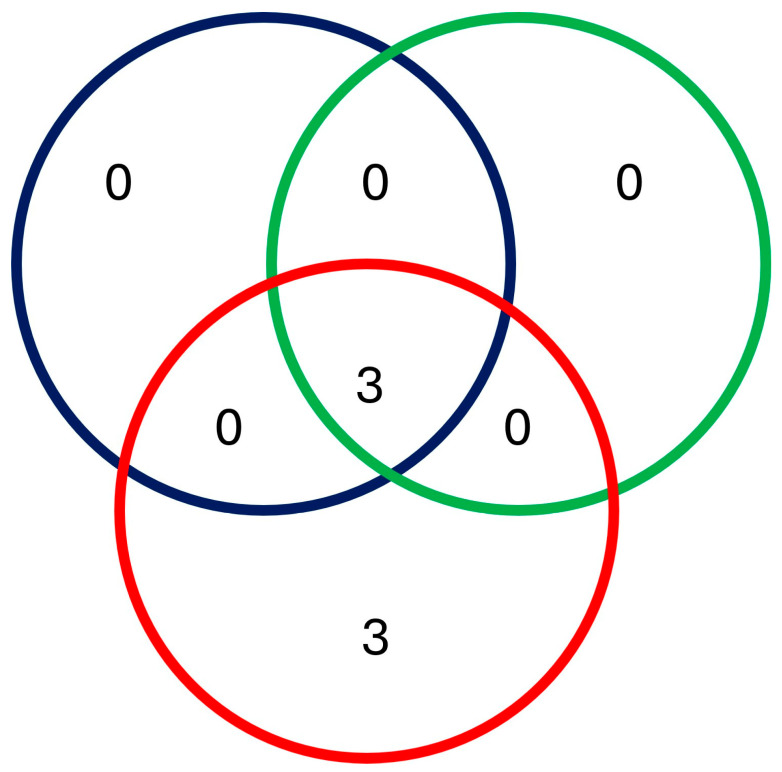
Venn diagram indicating shared *Stomoxys* species between communal, commercial, and game farm sites of north-eastern KwaZulu-Natal Province. Each setting is indicated by color: blue (communal farm), green (commercial farm), and red (game farm).

**Figure 5 insects-16-01049-f005:**
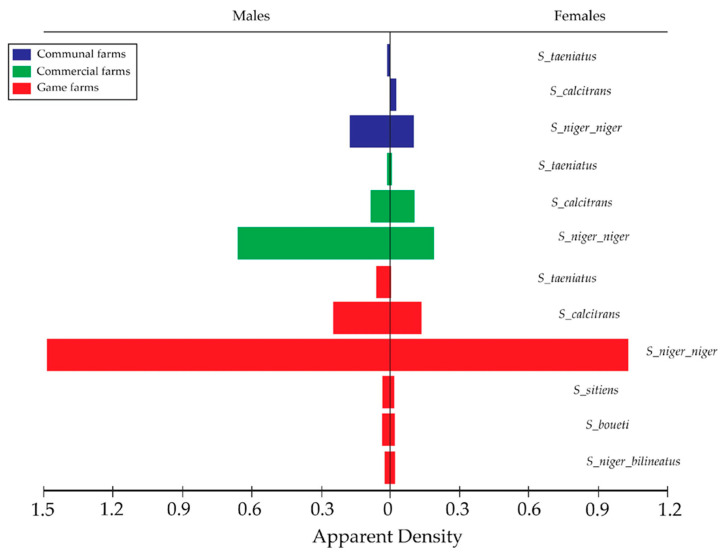
Funnel bars indicating apparent density (AD) of males and female *Stomoxys* flies for communal, commercial, and game farms in north-eastern KwaZulu-Natal Province.

**Table 1 insects-16-01049-t001:** The apparent density (AD) of *Stomoxys* flies collected from the three land-use setups.

Species_ID	Site_ID	No of Traps	No of Trapping Days	Male	Female	Total	Male_AD	Female_AD	Total_AD
	**Communal area**								
*S_calcitrans*	Ekphindisweni	3	22	1	2	3	0.02	0.03	0.05
*S_taeniatus*	Ekphindisweni	3	22	1	0	1	0.02	0.00	0.02
*S_niger niger*	Ekphindisweni	3	22	30	27	57	0.45	0.41	0.86
*S_niger niger*	Mvutshini	6	20	7	1	8	0.06	0.01	0.07
*S_niger niger*	Ocilwane	5	19	2	0	2	0.02	0.00	0.02
*S_calcitrans*	Tembe	3	14	0	1	1	0.00	0.02	0.02
*S_niger niger*	Tembe	3	14	13	2	15	0.31	0.05	0.36
	**Commercial Farm**								
*S_calcitrans*	Boomarang	5	23	7	7	14	0.06	0.06	0.12
*S_niger niger*	Boomarang	5	23	55	15	70	0.48	0.13	0.61
*S_calcitrans*	Silversand	1	7	6	0	6	0.86	0.00	0.86
*S_taeniatus*	Silversand	1	7	4	2	6	0.57	0.29	0.86
*S_niger niger*	Silversand	1	7	26	2	28	3.71	0.29	4.00
*S_calcitrans*	Sukari	2	7	10	24	34	0.71	1.71	2.43
*S_niger niger*	Sukari	2	7	112	39	151	8.00	2.79	10.79
	**Game Farm**								
*S_calcitrans*	Mkonge farm	4	7	0	9	9	0.00	0.32	0.32
*S_niger niger*	Mkonge farm	4	7	126	147	273	4.50	5.25	9.75
*S_niger bilineatus*	Mkonge farm	4	7	6	3	9	0.21	0.11	0.32
*S_sitiens*	Mkonge farm	4	7	9	3	12	0.32	0.11	0.43
*S_taeniatus*	Mkonge farm	4	7	6	0	6	0.21	0.00	0.21
*S_boueti*	Mkonge farm	4	7	5	3	8	0.18	0.11	0.29
*S_calcitrans*	Kuleni	3	33	2	1	3	0.02	0.01	0.03
*S_boueti*	Kuleni	3	33	3	0	3	0.03	0.00	0.03
*S_taeniatus*	Kuleni	3	33	5	1	6	0.05	0.01	0.06
*S_niger niger*	Kuleni	3	33	80	16	96	0.81	0.16	0.97
*S_calcitrans*	Zulucrock	2	33	71	30	101	1.08	0.45	1.53
*S_niger bilineatus*	Zulucrock	2	33	0	2	2	0.00	0.03	0.03
*S_niger niger*	Zulucrock	2	33	233	142	375	3.53	2.15	5.68
*S_sitiens*	Zulucrock	2	33	1	2	3	0.02	0.03	0.05
*S_taeniatus*	Zulucrock	2	33	4	0	4	0.06	0.00	0.06

## Data Availability

Voucher specimens are available at the ARC-OVR, and accession numbers for the sequences generated during the current study are available at the GeneBank.

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
