# Peer review of "Stomoxys Species Richness and Apparent Densities at Different Land-Use Setups in North-Eastern KwaZulu-Natal Province, South Africa"

_insects, 2025, doi:10.3390/insects16101049_

Round 1

Reviewer 1 Report

Comments and Suggestions for Authors

Introduction

  1. Reference 1 is focusing on Thailand, but it is used to back a statement of global scope. It is recommended that references of broader scope (ideally global) be used to back statements of global scope (such as reference 2). Reference 2 would be sufficient if no other reference of global scope is available.
  2. Reference 3 is focusing on Kenya, but it is used to back a statement of continental scope. It is suggested that a reference of continental scope be used.
  3. 58. It is recommended to replace “infectious diseases” with “infectious pathogens”, because the list that follows is not of diseases but of pathogens
  4. Reference 5 is focusing on Mali, but it is used to back a statement on the United States. It is recommended that the original source of the economic estimate be cited, not this paper indirectly reporting it.
  5. Ln 72. It is not clear why “feedlots and dairy farms” are in brackets, as if they were cases of “cattle markets”. Please check and revise.
  6. Ln 84. For consistency, it is preferable to cite the study Makhahlela et al., 2022 like all others (i.e. with a number), and not with the full name. The same applies to Evert (2014) further down.
  7. 91. KZN is first used here in the body of the text, so the acronym should be given in full.
  8. Ln 92. Replace “foci” with “focus”.
  9. Ln 91-92. The following references could be used to back the statement on the historical animal trypanosomosis (Nagana) focus
    • Kappmeier, K, Nevill, EM & Bagnall, RJ 1998, 'Review of tsetse flies and trypanosomosis in South Africa’. Onderstepoort Journal of Veterinary Research, vol. 65, no. 3, pp. 195-203.
    • De Beer et al. "An update of the tsetse fly (Diptera: Glossinidae) distribution and African animal trypanosomosis prevalence in north-eastern KwaZulu-Natal, South Africa." Onderstepoort Journal of Veterinary Research1 (2016): 1-10.
  10. Ln 92-93. The following reference could be used to back the statement on the southernmost distribution of tsetse flies (Glossina ) on the African continent.
    • Cecchi, G., Paone, M., de Gier, J. and Zhao, W. 2024. The continental atlas of the distribution of tsetse flies in Africa. PAAT Technical and Scientific Series, No. 12. Rome, FAO. https://doi.org/10.4060/cd2022en
  11. Ln 104. KZN acronym should be used, since it was introduced earlier.
  12. Ln 113. Please provide more details on how the study sites were chosen. Considering that the distribution of the sites is far from covering evenly the study area shown in Figure 1, certain criteria must have been used to define the exact location of the 10 study sites, and these criteria should be described here.
  13. Ln 121. The word ‘consortium’ could be removed.
  14. Ln 123. The authors may consider revising “by-catch”. For example, the Stomoxys flies may have been collected in surveys which had a dual purpose (sampling tsetse and other mechanical vectors of trypanosomosis), or that Stomoxys flies were opportunistically collected in sites where tsetse surveys was the main purpose. This description should be clarified also in the light of the comment above for Ln. 113, where more details on the selection of the study sites are requested.
  15. Ln 126. Please explain why a different number of traps was used in the different sites. Was it because the sampling was not designed specifically for the present study/Stomoxys but rather for general tsetse surveillance?
  16. Ln 142. The source of land use data are provided, but it is not explained exactly how land use attribute were extracted for the 10 study sites (if they were extracted at all). Was the land use of the sampling site used, or the land use of the broader area around the sampling site was considered?

Results

  • Ln 223. Please clarify the meaning of numbers such as ‘(107.5±89.5)’. Are these percentages? Can they be more than 100?
  • 229. Please clarify on how many flies/samples was the PCR analysis carried out. Was it done on all 1306 specimens? If not, how were the sub-samples selected? If possible, provide the number of samples tested by PCR for each species. And please clarify if these PCR-tested samples are all the same as those shown in Figure 2 for the sequencing.
  • Ln 255/Figure 3. The sites names in Figure 3 are not easy to read (especially for communal farms). Please try to improve the layout of the figure so that these are easier to read.
  • Ln 303. Table 1
    • Please consider renaming “Trap_days” as “No of days of trapping”; this is both for consistency with “No of traps” and for clarify. Alternatively, please consider adding the units of measurement in a new raw under the names of the columns (e.g. [No.] would denote “number” and [Flies/trap/day] would qualify the AD).
    • In the caption, consider removing “during the study period”, which is rather redundant, and rewording “settings” as “land-use setups” for consistency with the title. The latter recommendation for consistency should be considered throughout the manuscript; if the three types of areas you studied are defined as “land-use setups”, use this terminology consistently to avoid confusion. For example, in other points in the manuscript you refer to these types of areas as “ecological settings”.
  • Ln 307. It is not clear why the section “3.5. Land use in sampled areas” and the related Figure 6 are in the results section. The maps in the figure are not results of this study, and no analysis of land use from these maps and Stomoxys captures seem to have been conducted, so the maps are shown purely for descriptive purposes. It is therefore recommended that the maps be shown in the introduction or the materials sections (if they need to be shown at all), and that the sources of the maps be clearly indicated in the caption.
  • In addition to the point above, even without conducting specific analysis of land use/Stomoxys catches, it would be useful to briefly summarize what land uses are within (or near) the 3 “set-ups” analysed in this study (i.e. commercial, communal, game).
  • Finally, again linked to the point above, please try to clarify what you authors mean when you talk of “land use” in this paper. I believe you use the terms mostly for the 3 different “land-use setups” you studied, but sometimes also for the land use in Figure 6b. This inconsistent use of the term ‘land use’ creates confusion in the reader, and it should be clarified.

Discussion.

  • Ln 328-329. The text in these lines is not fitting well the ‘Discussion’ section. I believe it is suited for the Materials and methods section.
  • Ln 330-334. The text here is by and large a repetition of the results (and the same can probably be said for the whole paragraph in Ln 322-334). Please avoid repeating the same concepts in different places of the manuscript, and use the discussion for novel aspects not addressed already elsewhere.
  • Ln 348. Please reconsider the use of “however” as an opening to a new paragraph.
  • Ln 360. Please check the wording. What does “and until currently” mean?
  • Ln 363. Check the full stop before “suggesting” (I think it should be a comma)
  • Ln 370. Please check the wording. I do not think that “a large number of suitable hosts may be explained by the abundance of rotten organic materials”.
  • Ln 386. I think a “)” is needed after [44]
  • Ln 389. It is suggested to move to a new paragraph the text starting with “The higher species”, because otherwise the discussion on the types of traps and on the “environmental niches” is mixed up and can create some confusion.

Recommended points for further discussion

  • Given the uneven sampling of the KZN region for this study, how representative are the results of the overall regional (or district) diversity and abundance?
  • What other species (or genera/taxa) were captured in these surveys? Conceivably tsetse flies/Glossina, based on what explained in the methods, but perhaps also others, such as Tabanidae. More details on this would help place the study in a broader context of the diversity of disease vectors.
  • Is there any indication of the role Stomoxys currently play in the transmission of trypanosomosis in the study area?
  • For tsetse flies, continental and national-level mapping initiatives (i.e. the atlases) are now well established in several countries, including South Africa, and the same approach has also been recently adopted for potential mechanical vectors of trypanosomosis in Spain (see study below). What are the prospects and feasibility of applying the same approach for mapping Stomoxys at the national level in South Africa?
    • Melián-Henríquez et al. "Geographical distribution of potential mechanical vectors implicated in Surra transmission in Spain: an entomological perspective." Parasites & Vectors1 (2025): 305.

Conclusions

  • As they are now, the conclusions seem yet another summary of the study/results. Please try to focus conclusions on (1) future/planned research (if any), (2) envisaged concrete applications of the study results. For the latter, the text in Ln 411-413 is a start, but it is very vague and gives the impression that not much thought was put into it.

References

  1. Please ensure the use of italics for all species names.

Author Response

Most of the comments raised were corrected in the document and highlighted with a yellow highlighter; however, some were addressed as follows. 

Comments not addressed for Reviewer 1 are provided with more explanations below: 

  1. LN 126 The different numbers of traps used per different sampling sites was based on the availability of suitable tsetse habitat in the area as well as accessibility, to the sites. 
  2. Ln 142. Reply: Samples used were selected based on the availability of Stomoxys flies from the tsetse survey sites, as they were the main focus, and Stomoxys flies were other mechanical vectors that co-occured with the tsetse flies. 
  3. Ln 255. The images are too large to be used on the template provided, however, they can be shared on a separate document to be included by the journal. 
  4. Ln 307. The whole topic on land use was removed as suggested since not enough data was provided. 
  5. Ln 328 to 329. The paragraph was moved to the materials and methods section as suggested. 
  6. What are the prospects and feasibility of applying the same approach for mapping Stomoxys at the national level in South Africa? To generate a national atlas for Stomoxys flies, a wider survey to cover the whole of South Africa is needed, as we assume that different provinces, based on the different vegetation and climate, will have different species and their composition will also vary. It is, however, possible for such an atlas to be generated provided that ample funding is available. 

Reviewer 2 Report

Comments and Suggestions for Authors

Supplementary file 2 some pictures too dark, nothing is visible.

According to nomenclature rules, add authors of scientific names when first use it, probably here on line 24.

Line 54: Author of generic name Stomoxys  is not Zumpt but Geoffroy – correct this mistake.

Throughout text: if you recognize bilineatus as subspecies, please, name this as such and not as species. Moreover on line 222 you used bilineatus as species name.

Do not repeat in „keywords“ the same terms as are already in title of article.

  1. 65: Stomoxys are also known from boreal zones.
  2. 142 and 148: the same names of paragraphs but differently formatted(?)
  3. 235“Pharaphyly“ (?)
  4. 236 „calde“.
  5. 250. what is „0.0.57“.
  6. 253. „were“ not in capitals.
  7. 250 what is D value? (D is use at least in three different meanings in this paper).

From fig. 4 is is apparent that 4 species occur in communal farms but from fig. 5 it seems that only 3(?) And different numbers are put on line 246 („…while lower in all commercial and communal farm sites, recording only two species“).

What was source of data for fig. 6?

Numbers of references are sometimes omitted (e.,g., line 327), l. 346- dot between author and number?

  1. 386: „The ideal trap to be used for catching Stomoxyinae is the Vavoua trap [44].“ In what article you cited this statement? Surely not published in [44]. Probably Laveissiere, C. and P. Grebaut 1990?

Author Response

Comments raised by reviewer 2 are addressed in the attached manuscript and highlighted in a blue colour. 

Comment: Throughout text: if you recognize bilineatus as subspecies, please, name this as such and not as species. Moreover on line 222 you used bilineatus as species name.

Response: According to the study by Duvallet G, Hogsette JA. Global Diversity, Distribution, and Genetic Studies of Stable Flies (Stomoxys sp.). Diversity. 2023; 15(5):600. https://doi.org/10.3390/d15050600. These authors have reported that the two species (Stomoxys niger niger and S. n. bellinatus) are not sub-species based on genetic evidence, and they have suggested that a taxonomic revision is needed for the two species. As such, we have kept them as two separate species based on their findings. 

Comment: Supplementary file 2 some pictures too dark, nothing is visible

Response: Revised as suggested and a new Supplementary file added. 
